# Efficient and Shape-Sensitive Manipulation of Nanoparticles by Quasi-Bound States in the Continuum Modes in All-Dielectric Metasurfaces

**DOI:** 10.3390/mi15040437

**Published:** 2024-03-25

**Authors:** Lichao Zheng, Esha Maqbool, Zhanghua Han

**Affiliations:** Shandong Provincial Key Laboratory of Optics and Photonic Devices, Center of Light Manipulation and Applications, School of Physics and Electronics, Shandong Normal University, Jinan 250358, Chinaishach3137@gmail.com (E.M.)

**Keywords:** optical force, quasi-bound states in the continuum, all-dielectric metasurfaces, optical trapping, optical torque

## Abstract

Current optical tweezering techniques are actively employed in the manipulation of nanoparticles, e.g., biomedical cells. However, there is still huge room for improving the efficiency of manipulating multiple nanoparticles of the same composition but different shapes. In this study, we designed an array of high-index all-dielectric disk antennas, each with an asymmetric open slot for such applications. Compared with the plasmonic counterparts, this all-dielectric metasurface has no dissipation loss and, thus, circumvents the Joule heating problem of plasmonic antennas. Furthermore, the asymmetry-induced excitation of quasi-bound states in continuum (QBIC) mode with a low-power intensity (1 mW/µm^2^) incidence imposes an optical gradient force of −0.31 pN on 8 nm radius nanospheres, which is four orders of magnitude stronger than that provided by the Fano resonance in plasmonic antenna arrays, and three orders of magnitude stronger than that by the Mie resonance in the same metasurface without any slot, respectively. This asymmetry also leads to the generation of large optical moments. At the QBIC resonance wavelength, a value of 88.3 pN-nm will act on the nanorods to generate a rotational force along the direction within the disk surface but perpendicular to the slot. This will allow only nanospheres but prevent the nanorods from accurately entering into the slots, realizing effective sieving between the nanoparticles of the two shapes.

## 1. Introduction

It is known that light waves carry energy and linear and angular momenta [1]. For the nanoparticles (NPs) placed in the path of light propagation, the linear momenta will lead to optical gradient forces, while the angular momenta creates an optical torque, which causes the particle to rotate [2]. In 1986, Arthur Ashkin et al. proposed that a strongly focused laser beam passing through a high numerical aperture objective can form a stable three-dimensional optical potential well, formally known as a “single-beam gradient force potential well”, or optical tweezers for short [3]. Since then, optical tweezering technology, working as a contactless manipulation method [4], has been widely studied and implemented in biology and material sciences. There have been significant advancements in their practical applications, e.g., for trapping [5], releasing [6], and manipulating living cells or other biological particles [7]. Despite considerable progress, however, the trapping of subwavelength particles using a single focused laser beam is limited by the diffraction limit and, therefore, cannot trap nanoscale particles [8].

Although it is possible to capture and manipulate single micron-scale particles or living cells using conventional optical tweezering techniques, the capture of multiple particles, especially of those with submicron sizes and different shapes, remains a challenging task [9]. There are two major concerns in the manipulation of nanoparticles based on the optical approach. First, the level of used laser power is a key factor when trapping tiny particles. There is a relation between the size of the trapped particles and the required laser power [4], but this relation is not simply linear. Higher laser power is usually required to trap small particles that are less able to scatter and absorb light, so more light energy is needed to generate sufficient optical force. The laser light from higher-power lasers produces a thermal effect that can disrupt the structure of nanoscale particles or kill living cells. Second, current multiple-particle sorting methods, such as microfluidic centrifugal force [10], acoustic sorting [11,12], electric field-induced DEP [13,14], and electrophoresis [15,16], mainly focus on using differences in physical properties such as mass, size, refractive index, or conductivity between different particles to achieve effective sieving. However, sorting nanoparticles with similar refractive indices but different shapes is still a significant challenge. To address these issues, near-field light-capturing methods using plasmonic excitations have been actively investigated [17]. Plasmonic antenna arrays are able to break the diffraction limit in conventional optics by confining the electric field to the surface of metallic nanostructures, thus enabling particle trapping through the strong localization of near-field energy, which in turn enhances the optical gradient force [18]. However, the strong absorption loss of metals at optical frequencies due to the relatively large imaginary part of the metal permittivity causes the plasmonic nanoarrays to exhibit significant Joule heating, which usually decreases the stability of the trapping. This can cause various phenomena, such as the thermophoresis, convection, or boiling of water near the metasurfaces (where the trapping takes place), and even damage to the trapped specimens of particles. All of these factors limit the effectiveness of optical trapping techniques for certain biological cells or DNA chromosomes that are sensitive to temperatures, and further call for a nanoantenna array with lower loss and stronger field confinement. Recently, some emerging high-refractive-index semiconductor components have proven to be an effective alternative to metallic plasmonic materials in terms of modal confinement [19], and the dielectric material has the advantage of low loss, making it highly prospective for research on optical capture.

To address the high-power laser intensity problem, one needs to find an optical resonance that has the capability of providing large near-field enhancement but is free from absorption losses. In recent years, the bound state in the continuum (BIC), which refers to the non-radiation state of light with infinite quality factors, has received extensive attention in optics. By introducing proper perturbation, the BIC can be transformed into quasi-BIC (QBIC) modes with large quality factors and high field enhancement, which has attracted much attention in the field of optical trapping [20,21]. Our recent results suggest that an array of high-index disks with asymmetric open slots can support the high-Q resonance of QBIC modes [19], which provides a solid basis for its application in light capturing. Here, the QBIC resonance behaves as a super-cavity mode with a large quality factor, facilitating the requirement of only a low-power input to produce an ultra-large local electric field. So, this mode is particularly beneficial for the light capturing of NPs. It is known that the optical absorption in a structure can be characterized by $A=\int {0.5\omega \varepsilon^{”}|E|}^{2}dV$, where *ω* is the light angular frequency, *ε*″ is the imaginary part of the material permittivity, and |*E*| is the magnitude of the local electric field. Compared to plasmonic tweezers, the dielectric material should have *ε*″ at least five orders of magnitude (OFM) smaller while the |*E*| can be one OFM higher [19]. Considering the field volume of the QBIC supported by all-dielectric nanostructures to be additionally one OFM larger, one can estimate the Joule heating associated with the QBIC to be at least two OFM smaller than that by the plasmonic nanoantennas. We still use the AlGaAs disk antenna array on a quartz substrate to achieve a parallel capturing of multiple particles, and the Maxwell stress tensor (MST) technique is employed for the calculation of both the optical gradient force and the optical torque.

It is known that most biological cells or viruses in nature are spherical or rod-shaped. For example, spherical bacteria (*S. aureus*) and rod-shaped bacteria (*E. coli*) are two groups with a refractive index approximating 1.59, which will be used in our numerical simulations. They are widely found in serum and water and are capable of causing a wide range of human diseases [22]. Despite the crucial importance of classifying these differently shaped microorganisms to facilitate the study of pathogenesis, there is a lack of effective methods to separate homogeneous bacteria with different shapes [23] using techniques in the fields of microfluidics and acoustics [24]. Our designed structure mainly focuses on addressing this problem. To ensure feasibility in the fabrication of the structure, we set the dimension of the perturbation, i.e., the width of the slot *D* = 20 nm, which can result in a quality factor of the QBIC mode to the order of 10^5^. As schematically shown in Figure 1, an all-dielectric AlGaAs metasurface is designed on a quartz substrate, which supports the QBIC mode with a high-quality factor and significantly enhances the electric field. An aqueous solution containing spherical and rod-shaped particles is assumed to flow over the open-slotted disk array. The MST technique is used to calculate the optical force and torque on the nanoparticles at different locations above the open slot [25]. Our results show that within the QBIC modal field, both the nanospheres and the nanorods are subjected to high optical gradient force and optical torque, while the latter can be converted into rotational force. These two forces will pull the nanospheres towards the slots while rotating them around their own centers. In contrast, the rotational force will be along the long or short axis of the nanorods and tend to rotate them to be perpendicular to the slots. Since it is not possible for the nanorods to enter the open slot, the effective sorting of bacteria with different shapes [26] can be achieved. Whereas this work is based on numerical simulations, it shows that the optical force/momentum provided by the QBIC modes is superior for the shape-sensitive manipulation of nanoparticles. The results presented in this work may provide a guideline for future experimental studies. 

## 2. Results and Discussion

In this work, we design an all-dielectric metasurface with the capability of achieving multiple traps of the NPs simultaneously. We first explore the excitation of a QBIC mode supported by an array of slotted AlGaAs disks on a quartz substrate. To simplify the computational effort, the loss of the material itself is neglected when we are only interested in the narrow-band BIC phenomenon. Therefore, we set the refractive index of the AlGaAs and SiO_2_ to 3.2 and 1.444. The entire array is arranged into a square lattice with a periodicity of *P* = 900 nm. The dimensions of the disks include the radius *R* = 200 nm and height *T* = 450 nm. Each disk is etched through (see the inset in Figure 2b) by a slot with length *L* = 150 nm and width *W* = 20 nm. The center of the slot is laterally offset from the center of the disk by a distance of *D* = 20 nm. The original disk array without the slot supports a vertically aligned magnetic dipole (MD) mode that has perfectly circulated distributions of the electric field around the disk center and, thus, zero overlap with the field of the incident plane wave. In other words, it corresponds to a symmetry-protected BIC. By introducing the narrow slot inside the AlGaAs disk, the infinite-QBIC will be perturbed by the slot into the QBIC mode. The latter has a finite- yet high-quality factor controlled by the size and position of the slot and can be easily excited by a simple plane wave with linear polarization [19]. In addition, the QBIC mode field distribution, although slightly disturbed from that of the original BIC, still retains a similar profile. So, the excitation of the QBIC mode needs a plane wave with the polarization dependent on the direction of the slot. When the slot is along the *x*/*y* direction, the incident plane wave should be *y*/*x*-polarized. The transmittance spectrum through the structure has been numerically calculated by employing the finite element method (FEM) implemented in Comsol Multiphysics. In the calculations, Floquet periodic boundary conditions are used in the *xy* plane while perfectly matched layers are used in the *z* direction. Figure 2a shows the calculated transmission spectrum through the array in an aqueous (*n* = 1.33) environment, excited by a *y*-polarized plane wave incident from the top, for the structure with a slot along the *x* direction, as illustrated by the inset of Figure 2a. A broadband resonance around 1480 nm is seen, which is attributed to the excitation of a horizontally aligned MD mode with the moment along the *x* direction [27]. A much sharper resonance is seen on the right side of the MD resonance and a close-up of the sharp resonance is presented in Figure 2b. A high-quality factor of *Q* = 13,102 is found for this QBIC resonance, which is several orders of magnitude higher than that of the Fano resonance with an asymmetric line shape in plasmonic metasurfaces [28] or the Mie resonance in all-dielectric metasurfaces [29]. This high-Q resonance and the associated large local field enhancement make the QBIC mode a better choice for the capture and sieving of nanoparticles, as will be demonstrated in this work. 

The distribution of the electric and magnetic field amplitudes at the QBIC wavelength are normalized to the respective component of the incident plane wave and the results are presented in Figure 2c. The maximum of the electric field is localized in the interior of the slot and is enhanced by a factor of 93.8 compared to that of the incident plane wave which is assumed 1 V/m. Since the optical gradient force is proportional to the gradient of the local electric field, the electric field distribution explicitly gives a mapping of the optical force. The vectorial distribution of the displacement current given in the form of black arrows is also superimposed. Besides the well-confined mode distribution within the AlGaAs disk, it is also seen that a circulating displacement current is formed in the *xy* plane, indicating that it is an MD mode with the moment along the *z*-axis. The presence of the slot perturbs the electric field to be slightly removed from the symmetric state, allowing for the excitation of the mode in Figure 2c by an incident plane wave. To provide insights into the origin of the QBIC resonance, we further analyze the resonance, employing the multiple multipole decomposition technique [30]. This can be performed in the Cartesian coordinate system based on the displacement current using the following equation [31]:(1)$$J=i\omega \varepsilon_{0}(\varepsilon_{r}-1)E$$
where *ε*_0_ represents the vacuum dielectric constant and *E* is the total electric field inside the structure excited by the *y*-polarized plane wave. The calculated contributions from five different multipoles (MD (M), electric dipole (P), electric quadrupole (QE), magnetic quadrupole (QM), and toroidal dipole (T)) to the total scattered power are presented in Figure 2d. It is clear that at a QBIC wavelength of 1549.56 nm, the contribution from the MD resonance dominates, whereas the scattered power from the ED and TD is strongly suppressed. Also, one can observe that the MQ effect is slightly lower than the TD, while the higher-order EQ contribution is further suppressed. These results confirm that the QBIC resonance is mainly attributed to the MD excitation and also suggest that the sharp Fano-type transmission dip in Figure 2a is due to the coupling between two MD resonances, one vertical and the other horizontal.

It is known that nanoparticles may be trapped when they are in the vicinity of a highly confined mode due to large gradient optical forces generated by the evanescent field. The optical force can be characterized by the trapping potential. Alternatively, in the framework of classical electrodynamics, the component of the total time-averaged force F acting on an object can be calculated using the surface integral of electromagnetic fields combined with the MST technique [32]. The time-averaged optical force is calculated as
(2)$$F=\int_{S}\sigma \cdot n\;dS$$
where *S* is a closed surface surrounding the nanoparticle and *n* is the normal unit vector pointing to the external of the surface. The quantity *σ* is the time-averaged Maxwell stress tensor described as follows [33]:(3)σ=12Reε0ε1EE¯+μ0μ1HH¯−12ε0ε1EE¯+μ0μ1HH¯I
where *ε*_1_/*μ*_1_ and *ε*_0_/*μ*_0_ are the relative permittivity/permeability of the surrounding medium and the vacuum, respectively. Re is the operation to extract the real part, and *I* is the unit tensor. In addition, the optical torque *M* is induced by the surface traction force on the nanoparticles, and it can be calculated by the following [34]:(4)$$M\;=\;\int_{S}r\times (\sigma \cdot n)\;dS$$
where *r* is the position vector from a point on *S* to the center of the nanosphere.

The simulation of optical forces and moments was carried out using COMSOL Multiphysicss (Version 5.6), which specializes in meshing complex geometries and allows for an accurate treatment of electromagnetic fields on tiny particles. The obtained steady electromagnetic fields are used to calculate the optical force for the particles placed at certain positions above the disk. If the particles move or deform [35] due to the presence of optical force/moments, new calculations should be repeated to fully track the state of the particles. Figure 3a presents the optical forces in three directions (*F_x_*, *F_y_*, and *F_z_*) calculated with the MST technique for spherical nanoparticles (*r* = 8 nm) placed above the open slot at the incidence of a *y*-polarized plane wave with a low-power intensity (1 mW/µm^2^), where the inset illustrates a lateral shift between the centers of the sphere and the disk (*d* = 60 nm). The sign of the optical forces specifies their directions. When *F_x_*, *F_y_*, and *F_z_* > 0, the direction of the optical force is positive, which will push the particle along the direction of +*x*, +*y*, or +*z*. The calculation results reveal that at the excitation wavelength of the QBIC resonance, the optical forces on the spherical particles are dominated by the *F_x_* and *F_z_* components, where the maximum values of *F_x_* and *F_z_* reach 0.13 pN and −0.31 pN, respectively, while *F_y_* tends to be zero. In ref [4], a laser power intensity of 9.8 mW/μm^2^ was used to excite all-silicon nanoantennas to capture polystyrene spheres with a diameter of 20 nm, generating optical forces in the order of tens of fN, and the capturing of fluorescent particles was observed experimentally. In this paper, a lower power (1 mW/μm^2^) and similarly sized polystyrene spheres were used, yet with the produced optical forces one order of magnitude higher. So, the combined forces will try to pull the NPs toward the open slot. Because the *F_x_* and *F_z_* directions are different, a tremendous optical torque in the *y* direction will be generated, as shown in Figure 3b, where the optical torque magnitude upon the PS spheres is calculated. From the figure, we can see that *M_y_* has the largest value, which is consistent with our conjecture. When *M_y_* is positive, the spherical nanoparticles will produce a clockwise rotational force. Eventually, the spherical nanoparticles will rotate clockwise into the open slot. Compared with the optical torque in the *x* and *z* directions, the torque in the *y* direction is more significant. So, we only need to calculate the optical torque result in the *y* direction. In Figure 3c, we calculated the *M_y_* torque for PS spheres in aqueous solution at wavelengths from 1400 nm to 1700 nm. We find that the all-dielectric metasurface composed of asymmetric disks will produce a tremendous reverse optical torque at the Mie resonance and QBIC mode. The maximum value of *M_y_* is only −0.13 pN·nm at the Mie resonance wavelength of 1488 nm, while the maximum value of *M_y_* reaches up to 88.3 pN·nm at the QBIC mode, significantly facilitating the sieving of the particles. It is known that the magnitude of the MST is related to the electric and magnetic fields. From the distribution of the magnetic field in Figure 2c, one can see that it is not uniform. So, we further investigate the different optical force and torque when the PS spheres are put at different positions with respect to the center of the slot, by choosing two typical values of *d* to be 0 and 60 nm along the *x* direction. By comparing the results in Figure 3b,e, we can verify that the optical force on the particle becomes smaller as it moves away from the disk. In Figure 3d, the magnitude of the optical force on the PS spheres is shown, where the maximum values of *F_x_* and *F_z_* at *d* = 60 nm are 0.021 pN and −0.018 pN, respectively. When *d* = 0 nm, which means the PS sphere is at the center of the disk, these two values decrease to 0.0045 pN and −0.0011 pN, respectively. In Figure 3e, we present the calculated optical torque at these two different positions in Figure 3d, and the results show that the PS spheres are dominated by the *M_y_* torque in the band from 1549 nm to 1550 nm. The PS spheres will generate a clockwise rotational force positioned toward the center of the sphere. This rotational force is also reduced near the central disk position.

In light of the above study, we further investigate the optical trapping capability for another type of NP, the nanorods, at a QBIC wavelength of 1549.56 nm. The structure of the nanorods is schematically given in the inset of Figure 4a, where *l* = 30 nm, r = 8 nm, and *θ* indicates the angle between the midline of the long axis of the rods and the *x*-axis. We fix the position of the nanorods at *d* = 60 nm and *h* = 30 nm and rotate it along the *z*-axis while calculating the dependence of *M_y_* upon the rotation angle *θ*. It can be seen that the optical torque on the nanorods increases for larger angles because the MST integration surfaces become more oriented to the stronger field. In Figure 4b, it is seen that the optical force F_z_ first increases and then decreases as we set *θ* = 45°and *h* = 30 nm above the open slot while changing the position of the nanorod from −55 nm to 95 nm in the *x* direction. The corresponding optical torque is shown in Figure 4c, exhibiting a flipping of the direction, with a maximum value of −22.45 pN·nm in one direction and 48.22 pN·nm in the other. When the optical torque is negative, a counterclockwise rotational force will be generated, as shown in the inset, making the nanorods rotate counterclockwise, and as a result, the nanorods cannot enter the open slot. When the optical torque is zero near the center of the disk, no rotational force is generated. Therefore, under the same wavelength of linear polarized light, the nanorods placed at different positions experience different rotational forces. Combining this with the actual situation, we can boldly speculate that when the nanorods in the flow pass by the top of the open slot, the strongest clockwise or counterclockwise rotational force will be applied. Thus, they cannot enter the open slot, while the PS spheres are allowed to enter because of their shape. In this way, we can achieve our desired purpose of sieving: for the nanoparticles made from the same material but having different shapes.

## 3. Conclusions

As a conclusion, we present in this paper that an all-dielectric metasurface composed of circular disks with asymmetric open slots supporting QBIC mode can mitigate the Joule heat problem in plasmonic tweezers and provide high optical force and torque, facilitating the capture of nanoparticles of the same material with different shapes. By introducing void slots to the AlGaAs disk and optimizing the asymmetry of the slots, an experimentally feasible quality factor as high as *Q* = 13,102 can be obtained. We have demonstrated a very high electric field enhancement in the disk slots, which allows the capture of nanoscale particles close to the slots. We demonstrate that the high-quality factor and field enhancement of the QBIC mode can provide an optical force enhanced by two orders of magnitude over the plasmonic nanoarray along with a optical torque enhanced by three orders of magnitude over the Mie resonance at relatively low-power intensity (1 mW/μm^2^), allowing this system to finely sieve spherical and rod-shaped particles. We also demonstrate that the optical torque generated by rod-shaped particles at different positions above the open slot varies and acts as a rotational force in two different directions. This will substantially contribute to the further study of nanoparticle motion. This method of optical sieving nanoscale particles works well with limited power and does not require any complex structures. These efforts may provide additional sorting methods for particles of the same composition but different shapes, which can be used for virus screening and toxicity detection. And, this non-contact, wire-free, and frictionless actuation will also have other applications in micromachines, such as the precise positioning, gripping, and releasing of tiny objects and microfabrication, which bring many advantages in micromachine design and operation.

## Figures and Tables

**Figure 1 micromachines-15-00437-f001:**
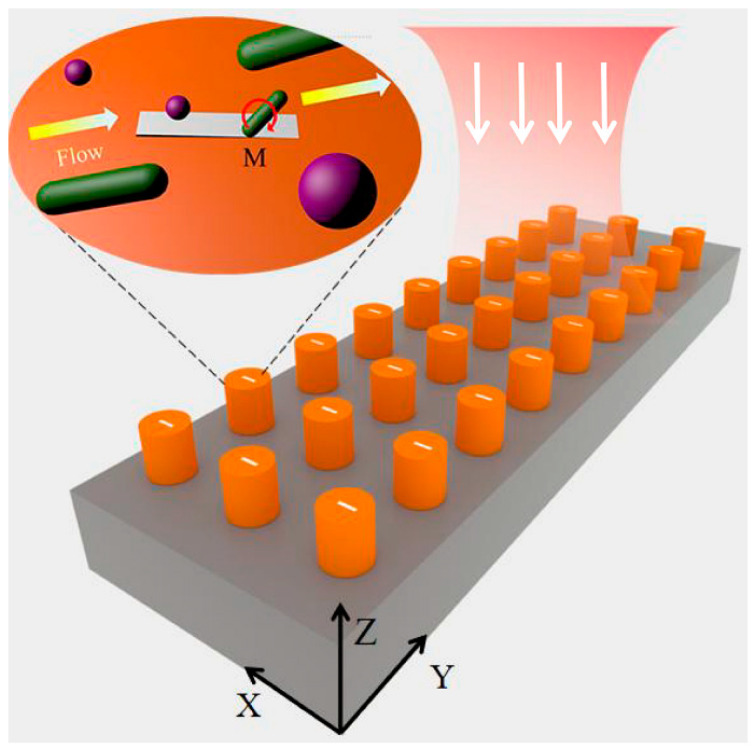
Schematic diagram illustrating the capturing of differently shaped NPs by the slotted AlGaAs disk array. The white, yellow, and red arrows indicate the incident plane wave, the direction of the solution flow, and the direction of the optical torque applied to the rod-shaped particles, respectively.

**Figure 2 micromachines-15-00437-f002:**
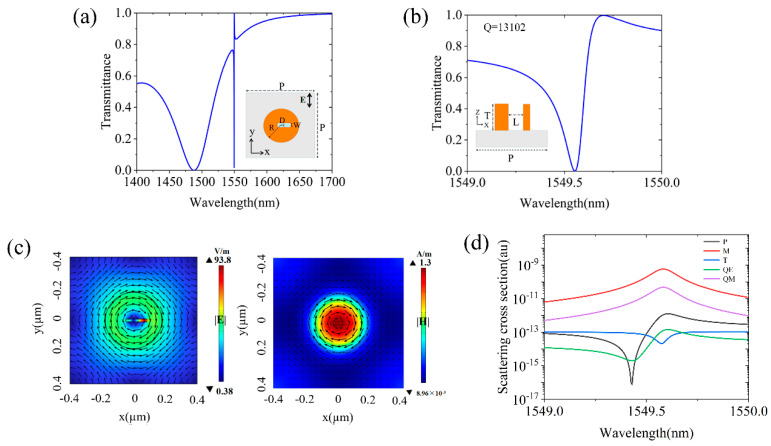
(**a**) Transmission spectrum through the slotted AlGaAs disk array under a *y* = polarized plane wave excitation. The inset shows the top view of one unit cell of the structure. (**b**) A close−up of the sharp resonance in Figure (**a**). The inset shows the side view of the slotted disk. (**c**) The amplitude of the electric and magnetic field as well as the vectorial distribution of the displacement current within one slotted disk at the QBIC resonance. (**d**) Multipole decomposition results of the total scattering power, with the logarithmic scale on the *y* = axis for a clearer comparison.

**Figure 3 micromachines-15-00437-f003:**
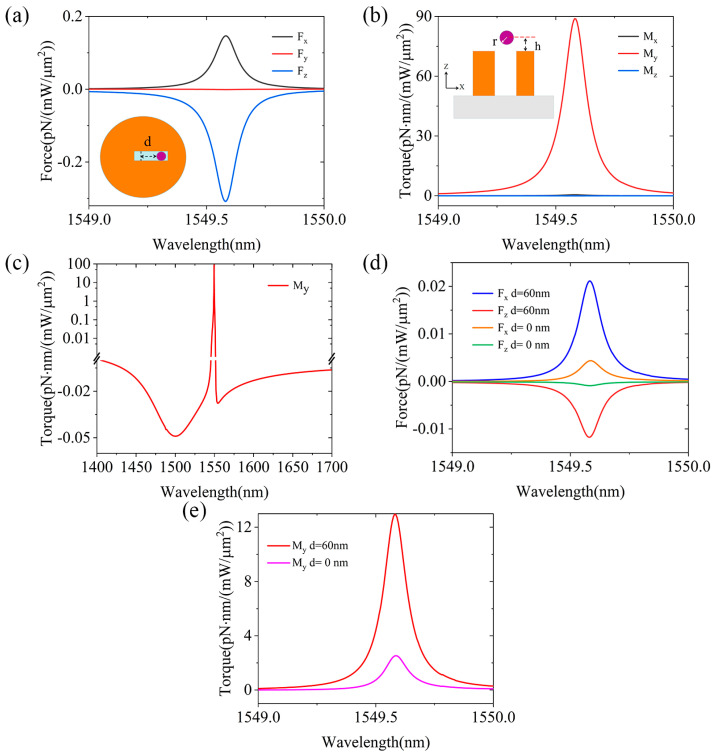
(**a**) The magnitude of the optical force upon the PS spheres on the disk (*d* = 60 nm). The inset indicates the top view of the sphere above the slot. (**b**) The magnitude of the calculated optical torque applied to the PS sphere over the open slot. The inset represents the side view of the sphere (*h* = 30 nm). (**c**) Calculated results of the change of *M_y_* on the PS sphere in the open slot. (**d**) Calculated results of the optical force on the PS sphere (*h* = 30 nm) at different distances. (**e**) Calculated magnitude of the optical torque from the results in Figure (**d**).

**Figure 4 micromachines-15-00437-f004:**
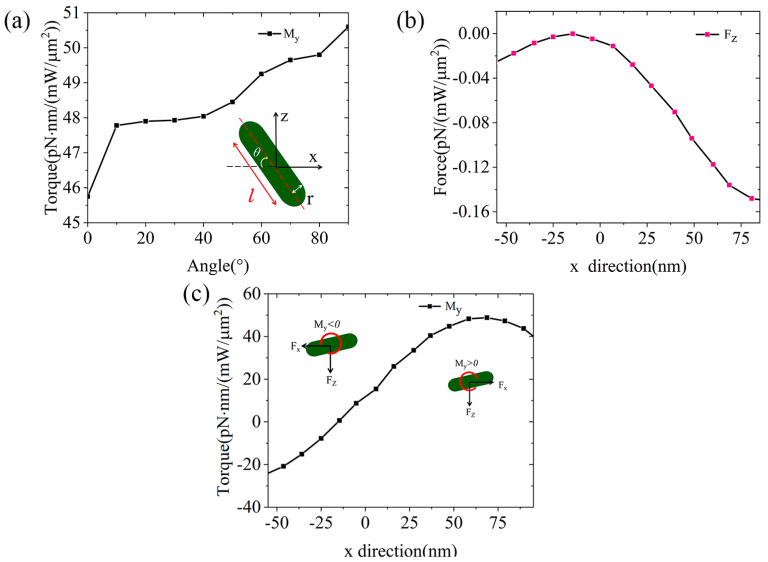
(**a**) The dependence of *M_y_* on the rotation angle of *θ* for the PS nanorods placed at the position of *d* = 60 nm, *h* = 30 nm; (**b**) the magnitude of the variation of *F_z_* with *x* position at *d* = 60 nm, *h* = 30 nm for the rod−shaped particles; (**c**) the magnitude of the optical torque for the PS nanorods at different position along the *x* direction.

## Data Availability

The original contributions presented in the study are included in the article, further inquiries can be directed to the corresponding author.

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
