# Peer review of "Efficient and Shape-Sensitive Manipulation of Nanoparticles by Quasi-Bound States in the Continuum Modes in All-Dielectric Metasurfaces"

_micromachines, 2024, doi:10.3390/mi15040437_

Round 1
Reviewer 1 Report
Comments and Suggestions for Authors
The authors proposed an array of high-index disks with asymmetric open slot that can support the high-Q resonance of quasi-bound states in the continuum (QBIC) modes, which provides a solid basis for its application in optical trapping. The simulation results are valid, however, a few problems still need to be addressed before it can be accepted.
1. Figure2a shows the transmission spectrum through the slotted AlGaAs disk array under a y-polarized plane wave excitation. I suggest the authors to mark the direction of y-polarization in the inset of Figure2a. Besides, why choose y-polarization and donnot consider the conditions of x-polarization?
2. From my perspective, the optical trapping force is directly proportional to the square of the electric field gradient, so I think the trapping results will be in accordance with the electric field distribution. However, the authors only demonstrated the relationship between the distribution of the magnetic field and optical force. The influence of electric field need to be further clarified with detailed data.
3. Normally,the trapping potential is used to characterize whether particles can be stably trapped. Therefore, I doubt if it is able to achieve an effective trapping. I suggest the authors to provide more concrete relevant data.
4. In Figure3a, the lateral shift between the centers of the sphere and the disk d =?
5. There are a lot of relevant research about BIC-assisted optical trapping, I suggest the authors supplement the references.
Comments on the Quality of English LanguageNO
Reviewer 2 Report
Comments and Suggestions for Authors
Comments
This manuscript introduces an interesting simulation work of shape-sensitive manipulation method for nanoparticles using optical-tweezer. Overall, the research is original and complete. It is recommended to be published after resolving a few comments as follows:
1. Could authors estimate the Joule heat level using their QBIC mode, and compare with the conventional optical tweezering technique?
2. Citations of the electric field-induced DEP method are missing in the Introduction (Line49-50), e.g.:
Matbaechi Ettehad, Honeyeh, et al. "Dielectrophoretic immobilization of yeast cells using CMOS integrated microfluidics." Micromachines 11.5 (2020): 501.
Qiang, Yuhao, et al. "Continuous cell sorting by dielectrophoresis in a straight microfluidic channel." ASME International Mechanical Engineering Congress and Exposition. Vol. 52026. American Society of Mechanical Engineers, 2018.
1. In addition to spherical and rod shapes, could authors also investigate other shapes, e.g., oblate, ellipse, etc.? A comprehensive investigation of the relationship between the shape-factor and optical torque/force will be very significant.
2. What was the MST method implemented? Is it in COMSOL or anything else? The authors should elaborate it in the section of Method. How was the tensors distribution across the surface of rod-shaped particles as opposed to the spherical particles? Will it reach an equilibrium position due to the torque? Will the initial position matter? Any dynamic process involved? A reference could be taken:
Qiang, Yuhao, et al. "Modeling erythrocyte electrodeformation in response to amplitude modulated electric waveforms." Scientific reports 8.1 (2018): 10224.
Comments on the Quality of English LanguageThe language needs to be carefully checked and improved.
Round 2
Reviewer 2 Report
Comments and Suggestions for Authors
The authors have addressed the reviewer's comments.
Comments on the Quality of English LanguageThe authors have addressed the reviewer's comments.